# Tobacco industry and public health responses to state and local efforts to end tobacco sales from 1969-2020

**Patricia A. McDaniel**[ID]*, **Ruth E. Malone**

Department of Social and Behavioral Sciences, School of Nursing, University of California San Francisco, San Francisco, CA, United States of America

* Patricia.McDaniel@ucsf.edu

## Abstract

### Background

In June 2019, Beverly Hills, California, became the first American city in the 21st century to pass an ordinance ending the sale of most tobacco products, including cigarettes, and it is unlikely to be the last. Knowledge of previous efforts to ban tobacco sales in the US, both successful and unsuccessful, may help inform tobacco control advocates' approach to future efforts.

### Methods

We retrieved and analyzed archival tobacco industry documents. We confirmed and supplemented information from the documents with news media coverage and publicly available state and local government materials, such as meeting minutes and staff reports, related to proposed bans.

### Results

We found 22 proposals to end the sale of cigarettes or tobacco products from 1969–2020 in the US. Proposals came from five states, twelve cities or towns, and one county. Most came from elected officials or boards of health, and were justified on public health grounds. In opposing tobacco sales bans, the tobacco industry employed no tactics or arguments that it did not also employ in campaigns against other tobacco control measures. Public health groups typically opposed sales ban proposals on the grounds that they were not evidence-based. This changed with Beverly Hills' 2019 proposal, with public health organizations supporting this and other California city proposals because of their likely positive health impacts. This support did not always translate into passage of local ordinances, as some city council members expressed reservations about the impact on small businesses.

### Conclusion

Tobacco control advocates are likely to encounter familiar tobacco industry tactics and arguments against tobacco sales ban proposals, and can rely on past experience and the results

**Data Availability Statement:** All relevant data are cited within the manuscript and/or Supporting Information files. The data we cite come from publicly available documents from the tobacco industry, the media, and state and local

governments. Our reference list includes hyperlinks to most of these documents. In three cases, the data (consisting of Board of Health and Town Council meeting minutes), while publicly available, were not posted on the world wide web. These data are contained in the Supporting Information files.

**Funding:** Funding for this work came from two awards granted to REM: award 26IR-0003 from the Tobacco Related Disease Research Program (http://www.trdrp.org/) and 19-10107 from the California Tobacco Control Program (https://www.cdph.ca.gov/Programs/CCDPHP/DCDIC/CTCB/Pages/CaliforniaTobaccoControlBranch.aspx). The funders had no role in study design, data collection and analysis, decision to publish, or preparation of the manuscript.

**Competing interests:** I have read the journal's policy and the authors of this manuscript have the following competing interests: PAM: Personal financial interests: Since 2006, I have been a full-time faculty employee of the University of California, San Francisco. My salary is provided by funds from research grants. I have received honoraria from the U.S. Department of Justice (for serving as a tobacco industry documents consultant for United States of America vs. Philip Morris, et al.), and Cancer Research UK (for preparing a report on views of the idea of a tobacco "endgame"). I have participated in tobacco control advocacy. Organizational financial interests: Within the last 5 years I have received grant funding from the California Tobacco-Related Disease Research Program (research funds derived from the state tax on cigarettes), and have worked on projects funded by the National Cancer Institute and California Tobacco Control Program. Interests of related parties: None to declare. REM: Personal financial interests: Since 1997, I have been a full-time faculty employee of the University of California, San Francisco. My salary is provided by funds from the state of California and my research grants. I have received travel/accommodation expenses and consulting fees or honoraria from the U.S. Department of Justice (for serving as a tobacco industry documents consultant for United States of America vs. Philip Morris, et al.), World Health Organization (for serving on the Expert Panel on Tobacco Industry Interference with Tobacco Control), American Legacy Foundation (for serving on an award selection panel), Clearway Minnesota (for serving as a grant proposal reviewer), U.S. Centers for Disease Control (for consulting on a tobacco industry documents research project), NIH (for serving as a grant proposal reviewer) and Cancer Research UK (for preparing a report on views of the idea of a tobacco "endgame"). I own

of a growing body of retail-related research to counter them. Considering how to overcome concerns about harming retailers will likely be vital if other jurisdictions are to succeed in ending tobacco sales.

## Introduction

In June 2019, Beverly Hills, California, became the first American city in the 21st century to pass an ordinance ending the sale of most tobacco products, including cigarettes, with sales scheduled to end in 2021 [1]. Manhattan Beach, California passed a similar ordinance in February 2020, with the same implementation date [2], and at least one other California city is exploring the idea [3]. Although public health groups have traditionally been wary of embracing any policy that might be framed as "prohibition," several supported these efforts [4, 5].

This was not the first time a US jurisdiction had ended the sale of cigarettes, the single most deadly consumer product in history. In the late 19th and early 20th centuries, spurred by concerns about youth smoking, the disease effects of cigarettes, and moral decline, 16 states banned cigarette sales, and 17 states considered doing so [6]. By 1927, however, under tobacco industry and public and media pressure, these laws were repealed [6, 7], and their history forgotten.

In the US, the current tobacco control movement began in earnest with the 1964 Surgeon General's Report [8]. More than fifty years later, smoking prevalence has declined dramatically, clean indoor air is widely accepted as the norm in public places and many private ones, and smoking has become denormalized [9]. The 2014 US Surgeon General's report called for an end to the tobacco epidemic, identifying several specific initiatives as particularly promising. Among them was the option for state or local jurisdictions to ban sales of whole classes of tobacco products [10, p. 856], as permitted by the 2009 US Family Smoking Prevention and Tobacco Control Act [11].

Ending tobacco sales could help smokers trying to quit, by removing environmental cues associated with smoking and decreasing cigarette availability [12–18]. It would send a strong message to the public that local government leaders were finally acting to protect public health by making policy that was consistent with messaging about the dangers of cigarette use [19]. Ending sales could also reduce youth tobacco uptake, which has been linked to tobacco outlet density near adolescents' homes [20], and tobacco use disparities, which have been linked to the greater concentration of tobacco outlets in economically deprived neighborhoods and in areas with greater proportions of African Americans and Hispanics [21–30]. Moreover, ending cigarette sales could further denormalize the tobacco industry, spurring additional reductions in cigarette uptake and use [31].

Any tobacco control policy innovation likely to be effective draws tobacco industry mobilization and opposition. A systematic review of strategies employed by the tobacco industry to influence tobacco taxes and marketing restrictions groups them into five broad categories: coalition management; information management; direct involvement and influence in policy; litigation; and illicit trade [32]. Each category encompasses more specific tobacco industry tactics, including using front groups to hide industry involvement, creating media or publicity campaigns to generate public support for the industry's position, commissioning research to support tobacco industry arguments, directly lobbying policymakers, working collaboratively (in the US, often through the now-defunct Tobacco Institute, a tobacco industry lobbying organization), and initiating legal challenges [33, 34].

one share each of Philip Morris International, Reynolds American, and Altria stock for research and advocacy purposes and have participated in tobacco control advocacy. I receive an annual honorarium and reimbursement of travel/ accommodation expenses from BMJ Publishing Group Ltd (for work as editor-in-chief of Tobacco Control). I have also received travel/ accommodation expenses and honoraria for speaking to various public health groups. In addition, I have received funding for reviewing documents as a potential expert witness for plaintiffs' legal cases involving tobacco industry activities. Organisational financial interests: Within the last 5 years I have received grant funding from the National Cancer Institute, National Institutes of Health, the California Tobacco-Related Disease Research Program (research funds derived from the state tax on cigarettes), and the California Tobacco Control Program. Non-financial interests: I have published or collaborated on research with more than 50 colleagues, postdoctoral fellows and students. I recuse myself from handling or reviewing papers submitted by these colleagues and others from my institution (UCSF). Interests of related parties: None to declare In 2019, both authors provided written informational resources at the request of Beverly Hills city staff and/or representatives and REM provided expert testimony at City Council or Health Commission meetings in Beverly Hills, Manhattan Beach, and Hermosa Beach. We confirm that nothing in those competing interest statements alters our adherence to PLOS ONE policies on sharing data and materials.

Research has also identified the arguments employed by the tobacco industry or its allies to influence tobacco control policies [33–35]. Many are used irrespective of the tobacco control measure in question, including the well-known and well-worn tobacco industry argument that adults should have the "freedom to choose" to smoke [36, 37], and the claims that tobacco control measures will hurt businesses and harm workers, spur illicit trade in tobacco products (thereby reducing government tax revenue and/or increasing consumption), or prove ineffective [38]. An examination of tobacco industry responses to novel tobacco control measures (e.g., a ban on tobacco additives, plain packaging) identified additional tobacco industry arguments, including those focused on the absence of and need for scientific evidence to support such measures [35]. Ulucanlar and colleagues have identified broad tobacco industry "discursive strategies" that encompass these and other tobacco industry arguments: exaggerating costs to the economy and society, public health, and the tobacco industry; exaggerating benefits to undeserving groups; and downplaying potential public health benefits [32]. Together, these strategies have the aim of creating "a dystopian narrative" that exaggerates the costs and denies or dismisses the benefits of proposed policies [32].

Tobacco company tactics and arguments in relation to tobacco sales bans have not been assessed. Given that such policies pose more of an existential threat to their business than other tobacco control policies such as tobacco taxes, tobacco companies might be expected to use new measures to oppose them. Anticipating any new tactics and/or arguments that tobacco companies may deploy may help tobacco control advocates counter them in future tobacco sales ban efforts. Alternatively, if tobacco companies are shown to rely on the same tactics and arguments when opposing sales bans as used when opposing other tobacco control measures, this knowledge could reassure advocates that they can build on their previous experience in countering tobacco industry opposition to tobacco control measures, rather than inventing a new approach.

The extent of efforts to enact sales bans prior to 2019 is also unknown. Before Beverly Hills' ordinance passed, we were aware from national news media reports of just two previous recent attempts to end tobacco sales in the US–one in Westminster, Massachusetts in 2014 [39], and one in the state of Hawaii in 2019 [40]–that both ended in failure. Other jurisdictions may have made similar efforts but attracted scant media attention. Knowledge of these efforts– including the tactics and arguments advanced by the tobacco industry or its allies and by public health organizations—may help shed light on how to account for, and replicate the success of the more recent ordinances. Understanding the prevalence of proposals to end sales may also help situate such proposals within a broader historical context.

This paper addresses the following research questions: 1) how many tobacco sales ban attempts have been made in the US since the 1930s, in which jurisdictions, when, and what was their scope?; 2) who initiated these efforts and why?; 3) what tactics did the tobacco industry employ to oppose these efforts, and were they the same or different from tactics previously identified in the literature?; 4) what arguments did the tobacco industry employ to oppose tobacco sales bans and were they the same or different from arguments previously identified in the literature?; 5) how did public health groups respond to efforts to end tobacco sales?; 6) what was the outcome of sales ban attempts?; and 7) are there any characteristics that distinguish successful from failed US efforts to end tobacco sales?

## Methods

For evidence of tobacco company tactics and arguments deployed in relation to tobacco sales ban proposals, we searched the Truth Tobacco Industry documents archive [41]. We relied on standard tobacco industry document search strategies [42–44], which include starting with

broad search terms and using retrieved documents to identify more specific terms, a process termed "snowball sampling." We used the initial search terms "sales ban," "ban sales of tobacco," and "ban tobacco sales." Retrieved documents allowed us to narrow our searches using the name of a particular state or locality, the name of persons or organizations involved in a particular sales ban proposal (e.g., Iroquois County Medical Society), the number assigned to a particular bill (e.g., Arkansas House Bill [HB] 467), and the year the proposal was introduced. We excluded items focused solely on military sales bans, vending machine sales bans, or youth access laws (sometimes discussed in the documents as bans on tobacco sales to youth). We identified 110 relevant documents dated from 1969–2000. We coded them for evidence of tactics and arguments used in relation to sales ban proposals by tobacco companies, tobacco industry allies, and public health organizations.

To confirm and supplement information retrieved from tobacco industry documents, we searched for news media coverage of any state or local proposal to ban cigarette or tobacco sales from 1969-March 2020. We used the online database Access World News, which indexes 1,607 US national and local news sources, including broadcast television news, talk radio, national public radio, newspapers and magazines, and web-only news sources. To locate news items concerning sales ban proposals identified in the tobacco industry documents database, we searched by year, name of city or state, any pertinent identifying information (such as the name of a legislator or organization proposing a sales ban), and the phrase "cigarette OR tobacco" AND "sales" AND "ban." To locate news items concerning any sales ban proposals not included in the tobacco industry documents database, we searched using the phrase "cigarette OR tobacco" AND "sales" AND "ban" during the period 1969-March 2020. We retrieved 109 news items, the majority concerning efforts to ban cigarette sales in 2014 and 2019. We also called government offices and searched government websites for publicly available material (e.g., bill language, staff reports or memos, committee or council agendas, minutes, video tapes of meetings, and written public comments) related to proposed bans in towns or cities in Illinois, Massachusetts, Maine, and California, and in the states of Arkansas, North Dakota, and Hawaii. We obtained material from the following towns, cities and states: Gray, Maine; North Adams, Salem, Worcester, and Westminster, Massachusetts; Elk Grove Village, Illinois; Beverly Hills, Carson, Hermosa Beach, and Manhattan Beach, California; and Arkansas, North Dakota, and Hawaii.

To develop this account, the first author reviewed all documents. Documents with an unclear provenance (e.g., those that were likely authored by a tobacco company, but lacking identifying information) or distribution (e.g., a Tobacco Institute press release that may never have been sent) were reviewed by both authors to solicit ideas for sources of corroborating evidence (n = 3). After reviewing material iteratively, we constructed a timeline of events, and organized and analyzed our findings in relation to our research questions.

## Results

Our tobacco industry documents and media searches located 22 proposals to end the sale of cigarettes or tobacco products from 1969–2020 (Table 1). These proposals came from five states (Arkansas, Massachusetts, Hawaii, North Dakota, and Utah), twelve cities or towns (the majority in Massachusetts and California) and one county (in Illinois) (Table 1). Most took the form of legislation introduced at the state level, ordinances proposed by city council members, or regulations considered by local boards of health (Massachusetts only); sales ban proposals originating from more "grassroots" sources, such as an individual (Worcester, Massachusetts 1998), or a tobacco control advocacy organization (North Dakota, 1990), were less common (Table 1). Proposals to ban all tobacco products, rather than cigarettes alone,

Table 1. Proposals to ban the sale of cigarettes or tobacco products, by year (1969–2020).

| Year | Location/Bill number | Legislation/Proposal | Who initiated/ proposed | Rationale/goal | Outcome |
|---|---|---|---|---|---|
| 1969 | Arkansas HB 467 | Bill to prohibit sale of tobacco products in any county unless approved by county voters (as done with alcohol) [45] | Representative David Kane | Alcohol and tobacco are equally harmful; this bill would bring the state's tobacco laws in line with its alcohol laws [46] | According to the Tobacco Institute, "the author of this Bill was persuaded to agree to its withdrawal because of the absurdity of trying to administer a prohibition Act" (sic) [47]. |
| 1970 1971 | Massachusetts HB 1544; HB 832 | Bills to prohibit the manufacture, distribution and sales of cigarettes in the state [45, 48, 49] | Representative James Nolen | To help smokers quit and to discourage youth smoking initiation [50] | Social Welfare Committee held hearings on both bills, recommending that the first be held over for the 1971 session and the second be rejected [6, p. lxviii]. |
| 1983 | Massachusetts | Legislation to ban the sale of cigarettes in state [51] | Public Health Commissioner Bailus Walker, Jr. | Inspired by the state's high cancer death rate to take an "aggressive approach" to a deadly product" [52] | Within 3 weeks, proposed legislation rejected by State Secretary of Human Services and Secretary of Administration and Finance [53]. |
| 1990 | Hawaii SB 2209 and HB 2249 | Bill to prohibit the sale of tobacco products in the state after December 31, 1999 [54, 55] | Senator Russell Blair, chair of Consumer Protection Committee | To "prevent the next generation of tobacco addicts from being seduced by the murderous tobacco industry."[54] | Bill died in committee [56]. |
| 1990 | North Dakota | Plan to work towards legislation to ban tobacco sales in the state by January 1, 2000 [57, 58] | Tobacco Free North Dakota and North Dakota Cancer Coalition, with help from the state health department's Cancer Prevention and Control project | Tobacco is a deadly product with "no useful purpose" [58] | No bill materialized. |
| 1990 | Iroquois County, IL | Proposal to ban the sale and use of commercially made cigarettes in the county by the year 2000 | County Medical Society | To help smokers quit, improve health, and lower medical costs [59] | County officials "refused to consider a ban," as did officials in the county's largest city [60]. |
| 1994 | Salem, MA | In a discussion focused on regulating cigarette vending machines, a proposal emerged for an outright ban on cigarette sales in Salem [61] | Board of Health member Owen Meegan | Unknown | The proposal appeared to not be serious; the Board focused instead on creating regulations governing cigarette vending machine sales [62]. |
| 1996 | North Adams, MA | Proposal to ban tobacco sales, tobacco advertising, and smoking in all public places [63] | City Council member Donovan | City Council member cited as precedent a previous ban on a plastic, aerosolized toy (silly string) "deemed bad for the public welfare." [64] | After a City Council meeting attended by more than 50 restaurant owners and retailers, with no speakers voicing support of the proposal, the council voted 7–1 (with 1 abstention) not to move forward with proposalp [65]. |
| 1997 | Winthrop, MA | Proposal to ban sale of tobacco products [66, 67] (Would have affected 25 existing retailers) | Board of Health, backed by 3-member Board of Selectmen | To reduce youth smoking and the number of smoking-caused deaths [68] | Over 6 months, the Board sought support from neighboring towns and state legislators [67], hosted a "raucous" public hearing (attended by 200–400 residents, most opposed to the proposal) [69–71], and requested permission for a non-binding ballot measure assessing community support for a sales ban [72]. Six months after the proposal's introduction, the Board of Selectmen withdrew its support and the Board of Health abandoned plan [73, 74]. |
| 1998 | Gray, ME | Proposal to ban sale of tobacco products [75, 76] | Town Council Chair Mark Sanborn | To reduce youth smoking [75, 76] | Town Council voted 3–2 against holding public hearing on proposal [77]. Two Councilors who were opposed to the public hearing expressed opposition to "taking individual rights away from people" with a tobacco sales ban [78]. |
| 1998 | Utah | Amendment to ban sale of tobacco products added to a bill banning self-service tobacco displaysl [79, 80] | House minority leader Dave Jones | Spur of the moment proposal: "We were arguing over where to put the tobacco and it donned [sic] on me that we were just pretending to deal with the problem."[79] | A mix of Democratic and conservative Republican lawmakers "who hate tobacco use [81] initially approved the amendment, 45–11 (with 19 abstentions) [82]. After Republican leaders "rallied" their fellow Republicans, amendment was dropped from the bill within hours [79, 81]. |

(*Continued*)

**Table 1.** (Continued)

| Year | Location/Bill number | Legislation/Proposal | Who initiated/ proposed | Rationale/goal | Outcome |
|---|---|---|---|---|---|
| 19982000 | Worcester, MA | Citizen petition requesting that the City Council consider prohibiting tobacco sales [83, 84] | Mayor's 10 year old son, Anthony Mariano | Unknown | 1998: Public Health Committee heard the request, but took no action [85]. |
| | | | | | 2000: City Council agreed to consider raising legal purchasing age for tobacco to 21 years of age instead of ban [86]. Public Health Committee sought information from administrators on how to increase the purchasing age [87]. |
| 2003 | North Dakota HB 1174 | Bill to prohibit sales and use of tobacco products | Representative Mike Grosz | Economic and health consequences of smoking; a "means to save lives" [88]. (Some public health groups believed that the real goal was to make them look bad, in order to reduce support for a proposed cigarette tax increase [89]). | House Finance and Taxation Committee held a hearing on this bill, with testimony from public health organizations [90]. The Committee recommended passage, 9–4, reportedly out of frustration with health groups' opposition to the bill [90]. Bill ultimately voted down, 88–4 [89]. |
| 2006 | Elk Grove Village, IL | Proposal to ban the sale of tobacco products | Mayor Craig Johnson | Responding to a proposal to ban smoking in public places, the mayor suggested that a sales ban would better protect public health) [91] | Two months later, Village Board agreed to abandon the sales ban proposal in favor of a new fee structure for tobacco retail licenses, with all revenues earmarked for smoking cessation programs [92]. |
| 2014 | Westminster, MA | Board of Health proposal to ban sale of tobacco products (would have affected 8 retailers) [93] | Board of Health | Board members saw it as more fair than a proposal to only ban tobacco sales in pharmacies, and a way to "protect the health of residents in Westminster" [94]. | Over 8-month period, Board of Health sought input from local government officials and from residents and businesses. 200–600 residents attended Board of Health hearing, which was shut down by the Board after 23 minutes, due to audience behavior [93, 95–97]; police escorted Board members from the building [93]. Board of Selectmen and the Economic Development Committee stated their opposition to the proposal, and town residents initiated recall effort for 2 members of the Board of Health [98]. Board voted 2–1 to drop proposal, [99] and the recall effort was abandoned [100]. |
| 2017 | Elk Grove Village, IL | Proposal to ban sale of tobacco products | Mayor Craig Johnson | When resident proposed raising age of purchase to 21, Mayor proposed to "stop nibbling at the edges. Let's do this right and ban the sale of tobacco completely" [101]. Village shouldn't sell product "guaranteed to harm residents" [102]. | One month later, the mayor withdrew the proposal, citing the costs of likely litigation, and the possibility of state preemptive action [101]. |
| 2018 | Saratoga, CA | Proposal to ban sale of all tobacco products | City Council member Howard Miller | Alternative to ban on sales of flavored tobacco products. Miller saw sales bans as inevitable. Saratoga could be first, with other cities "catch[ing] up to us for a change" [103]. | The City Council directed staff to prepare an ordinance for review; the city also planned to contact retailers for their input [103]. Two months later, the City Council directed staff to prepare ordinance to prohibit sales of flavored tobacco instead of prohibition on all sales [104]. |
| 2019 | Hawaii HB 1509 | Bill to ban cigarette sales by increasing minimum age of purchase over 5 year period [105]. | Representative Richard Creagan, M.D. | State obligated to "protect the public's health" [106] | The bill was referred to the House Committee on Health, where Representative Creagan proposed two amendments: 1) to raise the legal age to purchase tobacco products from 21 to 25; 2) to solicit a study from the Hawaii State auditor on the implications of his original proposal [107]. Even with these changes, all committee members voted to hold the bill indefinitely, effectively killing it [108]. The committee chair stated that "The state of Hawaii's just not ready for this massive change yet" [107]. |

*(Continued)*

**Table 1.** (Continued)

| Year | Location/Bill number | Legislation/Proposal | Who initiated/ proposed | Rationale/goal | Outcome |
|---|---|---|---|---|---|
| 2019 | Beverly Hills, CA | Proposal to ban sale of tobacco products (including electronic cigarettes), effective January 1, 2021 (will affect 17 retailers; 3 existing cigar lounges and concierge sales to hotel guests [currently offered at 8 hotels] are exempted) [109] | Mayor Julian Gold and Vice Mayor John Mirisch | Health and economic consequences of smoking; desire to reduce youth smoking initiation [110] | Over 6-month period, the Health and Safety Commission met 4 times to discuss issue [111], and city staff wrote a report [109], gathered feedback from tobacco retailers and other stakeholders, including public health organizations [112], and wrote a draft ordinance [110]; it also identified resources, such as consultations with the Small Business Development Center, that retailers could draw on to help with the transition. The City Council made some modifications to the proposed ordinance (e.g., date of implementation, hotel exemption) [113]. Ordinance had its first reading on May 21, 2019, and passed unanimously at its second reading on June 4, 2019. |
| 2019 | Carson, CA | Proposal to ban sale of tobacco products, including electronic cigarettes (would affect 72 retailers) | Mayor Albert Robles | Beverly Hills' action made mayor aware that "it was possible to do this" [114] to protect public health. | City Council voted 4–1 to table the motion. One City Council member objected to the proposal on the grounds that it would not reduce smoking prevalence and would hurt local businesses. He also noted that retailers had not been informed, and that he had not seen the proposal before it was introduced [114]. |
| 2019 | Hermosa Beach, CA | Proposal to consider ban on tobacco and electronic cigarette sales (would affect 14 retailers) | City councilmember Jeff Duclos | To keep pace with Manhattan Beach (see below) | In October 2019, council member Duclos requested a future agenda item to consider a tobacco and e-cigarette sales ban [115]. In November 2019, city council discussed potential ban, and directed staff to return with options for prohibiting the issuance of new tobacco retail licenses, banning the sale of all vape products, and facilitating discussion with retailers about a process and time frame for ending all tobacco sales [116]. In January 2020, council met to discuss a) an ordinance to stop issuing new tobacco retail licenses and to prohibit the sales of electronic smoking devices; and b) whether to consider the process and timing of a future ban on tobacco sales [117]. The City Council agreed 4–1 to pass the ordinance prohibiting new tobacco retail licenses and the sale of electronic cigarette devices. Two city council members expressed support for proceeding with a ban on tobacco sales, but 3 had reservations, focused on negative impact on retailers. The council voted to re-visit the idea with staff in June 2021, 6 months after the Beverly Hills and Manhattan Beach ordinances had been in effect [117]. |

(*Continued*)

**Table 1.** (Continued)

| Year | Location/Bill number | Legislation/Proposal | Who initiated/ proposed | Rationale/goal | Outcome |
|---|---|---|---|---|---|
| 2020 | Manhattan Beach, CA | Proposal to ban sale of tobacco products (including electronic cigarettes), effective Jan 1, 2021 (will affect 18 retailers) [118] | Mayor Steve Napolitano | Health and economic consequences of smoking [119] | In October 2019, City Council met to discuss 3 possible restrictions on tobacco sales: ban on sale of flavored tobacco products (with non-flavored tobacco products and electronic cigarettes sold in adult-only stores); ban on sale of electronic cigarettes and all flavored tobacco products (with non-flavored tobacco products sold in adult-only stores); ban on sale of all tobacco products and electronic cigarettes [120]. Council voted 4–1 to ask city staff to return with 1) an urgency ordinance that would immediately ban the sale of all electronic cigarette products and flavored tobacco and 2) an ordinance that would prohibit the sale of all tobacco products in the near future [121]. In November 2019, City Council adopted urgency ordinance to immediately ban sale of all flavored tobacco products and electronic cigarettes [122]. City staff gathered feedback from retailers and public health experts on sales ban proposal, and identified LA County resources to help retailers transition [118]. In December 2019, city staff recommended adopting an ordinance banning the sale of all tobacco products. In February 2020, the City Council voted 4–1 to pass the ordinance, with the member who opposed it citing concern for a ban's impact on retailers [2]. |

Abbreviations: CA California; IL Illinois; MA Massachusetts; ME Maine.

were more common (n = 17), and the most recent proposals (2019 and 2020) included both e-cigarettes and traditional tobacco products (Table 1). Proposed cigarette-only bans occurred primarily between 1970 and 1994, with the 2019 Hawaii proposal the lone exception (Table 1). Only one proposal (North Dakota HB 1174, 2003) prohibited both sales and use of tobacco products; the remainder were focused solely on sales (Table 1).

Individuals or organizations proposing sales bans typically justified them in the name of public health, citing the health harms associated with tobacco product use, and/or the likelihood of a ban facilitating smoking cessation or reducing youth smoking. Alleviating the economic burden of tobacco use was also occasionally mentioned as a justification. Rationales for these proposals were typically dispassionate, with the exception of Hawaii's SB 2209 (1990) which was framed as "prevent[ing] the next generation of tobacco addicts from being seduced by the murderous tobacco industry" (Table 1). Between 1998 and 2018, several sales ban proposals emerged as spontaneous responses to other tobacco policy proposals, such as banning

the sale of tobacco in pharmacies, or raising the legal age of sale from 18 to 21, with the sales ban proposal offered as an alternative that was more fair to retailers (Westminster, MA 2014), more comprehensive (Utah 1998), more effective in protecting public health (Elk Grove Village, IL 2006), or, in one case, inevitable, giving the city the opportunity to take the lead on this issue (Saratoga, CA 2018).

It was only recently that any of these proposals garnered enough policymaker support to be adopted, with two Southern California cities (Beverly Hills and Manhattan Beach) passing tobacco sales ban ordinances in 2019 and 2020 (Table 1). Some proposals failed almost immediately, providing little opportunity for the tobacco industry or public health organizations to respond (Table 1). For example, a formal proposal to ban the sale of tobacco products in Utah in 1998—a "spur of the moment" response to a discussion of a self-service tobacco display ban that initially garnered support from lawmakers–was ultimately defeated several hours after its introduction (Table 1). Other proposals were considered and debated by policymakers and the public over the course of weeks or months, giving the tobacco industry and public health organizations more time to act. In several cases, it appeared that opposition from (or concerns about) retailers and the local community played a significant role in policymakers' decision not to ban tobacco sales (e.g., North Adams, Winthrop and Westminster, MA; Hermosa Beach, CA) (Table 1).

The Tobacco Institute took the lead on monitoring and coordinating tobacco industry responses to early attempts to ban tobacco sales. It was somewhat dismissive of these attempts, describing them internally as "absurd" (Arkansas 1969) [47], "strange" (Massachusetts 1970) [123] and "outlandish" (Hawaii 1990) [124], with only a "remote" chance of succeeding (Massachusetts 1983) [125]. Nonetheless, the Institute vowed not to take such efforts "lightly" [125]. This approach was evident even in cases where the proposal to ban cigarette sales appeared not to be serious. For example, in 1994, in Salem, Massachusetts, the Board of Health discussed over several months how best to regulate vending machine cigarette sales [62]. In one early meeting on the topic, a Board Member suggested exploring an outright ban on cigarette sales in Salem. Although this idea was not raised in subsequent meetings, the Tobacco Institute was aware of the proposal, mentioning it in its regular newsletter on state tobacco control activities [126]; the Institute may also have been responsible for preparing a document (with no assigned author) entitled "In Opposition to a Cigarette Sales Ban" outlining, over 6 pages, why Salem's "ill-founded" proposal should be rejected [127].

The Tobacco Institute and individual tobacco companies relied on few tactics to oppose proposals to ban sales– 4 of over 20 tactics identified in the literature on tobacco industry efforts to influence tobacco taxes and marketing restrictions [33, 34] (Table 2). Two were captured under the broader category of coalition management [32]: tobacco companies working collaboratively, often under the auspices of the Tobacco Institute, and mobilizing allies, such as retailers associations, to oppose tobacco sales bans (Table 2). For example, in response to proposals by two Massachusetts towns to ban tobacco sales in 1997, tobacco companies encouraged restaurant owners, retailers, smokers and residents to express their opposition at public hearings [64, 128] (Table 2). In one case, over 50 "partisans and coalition partners" showed up and spoke out against the proposal in the public comment period [129]; in the other, 200–400 reportedly attended a "raucous" public hearing, most of them opposed to the proposal [69–71] (Table 2). Two other tobacco industry tactics were captured under the broader category of information management [32]: commissioning supportive economic and legal research, and creating media or publicity campaigns, including letters to the editor and press releases. (Table 2). In 1990 in Hawaii, for example, the Tobacco Institute commissioned both a legal analysis (from the law firm Covington and Burling) and an economic analysis

(from Price Waterhouse) to aid in its opposition to a proposed tobacco sales ban (Table 2). We found evidence of no new tactics.

The most common tactics were working collaboratively and mobilizing allies, such as retailers' associations, to oppose sales ban proposals in public meetings and via petitions to policymakers. We found no evidence of direct tobacco company involvement in policy via lobbying, or threats of litigation, although a law firm representing a local retailer sent a letter to the Manhattan Beach City Council in 2019 threatening to sue [146]. Hawaii's 1990 proposal to end tobacco sales by the year 2000 saw the most high profile involvement by the tobacco industry, featuring a variety of tactics organized by and publicly identified with the Tobacco Institute. Following that, the tobacco industry worked primarily behind the scenes, through allies.

Table 3 summarizes the specific arguments that the tobacco industry and its allies (e.g., retailers associations, Chambers of Commerce) relied on to oppose tobacco sales ban proposals. These arguments encompassed the discursive themes previously identified in the literature: an emphasis costs to society, the economy, law enforcement, and the tobacco industry, and the denial of intended public health benefits [32]. Within these categories, none of the specific arguments were unique to tobacco sales ban proposals. For example, the argument that a sales ban in Beverly Hills, California would foster antagonism against store clerks by irate customers was previously used to oppose plain packaging legislation in New Zealand [147]. Similarly, the claim that a particular health body (i.e., Salem, Massachusetts' Board of Health) lacked the authority to impose a sales ban was also used to oppose a proposed ban on tobacco additives by Brazil's National Health Surveillance Agency in 2012 [148]. Even arguments that might conceivably be reserved for proposals to ban cigarette or tobacco sales–namely, that such proposals were uniquely extreme or unprecedented–were also used in other policy arenas, including early battles over clean indoor air [149]. Some tobacco industry arguments were used less often overall than others, including those focused on the economic costs to governments, and, among arguments focused on societal costs, claims of a "slippery slope" and an absence of legal authority to ban tobacco sales (Table 3). Among the most common tobacco industry arguments to oppose sales bans were those focused on economic costs to the tobacco industry (specifically, retailers) and the creation of black markets (with references to alcohol prohibition in the US); both arguments were also used regularly from the 1990s onwards (Table 3). The claim that sales bans would not reduce smoking was used more often in the most recent efforts to ban tobacco sales.

We were unable to identify public health organizations' positions on and activities related to all 22 tobacco sales ban proposals; however, from the limited evidence available, certain patterns emerged (Table 4). Before Beverly Hills, public health groups who took a position on sales ban proposals were usually opposed; in the few cases where a public health organization expressed support (Winthrop, MA 1997; Elk Grove, IL 2006), it was muted. In a newspaper article, a representative of the American Cancer Society praised the Winthrop Board of Health's proposal to ban tobacco sales, but offered no other support, such as speaking at a public hearing (Table 4). In Elk Grove, two representatives of the American Lung Association (ALA) did speak at a public hearing, but gave a mixed message, stating that while the ALA did not regard a tobacco sales ban as a policy priority, it would support it if passed (Table 4). The primary rationale for opposing cigarette or tobacco sales bans offered by public health organizations was that there was no evidence demonstrating sales bans' effectiveness in reducing smoking prevalence. Instead, public health organizations recommended focusing on proven tobacco control strategies (Table 4).

Beginning with Beverly Hills' proposal, public health organizations' attitudes changed, with numerous organizations expressing strong support for tobacco (and electronic cigarette) sales bans in Beverly Hills, Hermosa Beach, and Manhattan Beach (Table 4). The rationales for this

**Table 2. Tobacco industry strategies and tactics to oppose proposals to ban cigarette/tobacco sales.**

| Strategy | Specific Tactic | Where (when) employed | Examples |
|---|---|---|---|
| Coalition management | | | |
| | Work collaboratively, primarily through the Tobacco Institute | MA (1970/1971, 1983) | • MA (1970/1971, 1983): The Tobacco Institute took the lead on monitoring these bills [123, 130].<br>• HI (1990): The Tobacco Institute organized the media, third-party, and consultant response to the proposed bill to end tobacco product sales [131–137] |
| | | HI (1990) | |
| | | Salem, MA (1994) | |
| | | North Adams, MA (1997) | |
| | | Worcester, MA (1998, 2000) | |
| | Mobilize allies (e.g., retailers associations, smokers, Chambers of Commerce) | HI (1990) | • North Adams, MA (1997): Philip Morris and RJ Reynolds encouraged restaurant owners, retailers and smokers to speak at a public hearing in opposition to a proposal to ban tobacco sales [64]; over 50 tobacco industry "partisans and coalition partners" showed up and many spoke out against the proposal in the public comment period [129].<br>• Winthrop, MA (1997): Philip Morris USA contacted smokers by phone to inform them of the Board of Health proposal and encourage them to attend a public hearing on the matter, seeking those who were "articulate" and "strongly opposed" [128]. 200–400 people attended the hearing; most opposed the proposal [69]. Retailers also presented board with 2,600 signature petition against proposal [68].<br>• Westminster, MA (2014): New England Convenience Store Association supported signature gathering for petition opposing ban [138]. It was presented at the Board of Selectmen meeting (with 1,000 signatures) [139]. |
| | | North Adams, MA (1997) | |
| | | Winthrop, MA (1997) | |
| | | Gray, ME (1998) | |
| | | Westminster, MA (2014) | |
| Information management | | | |
| | Commission supportive research | HI (1990) | • HI (1990): The Tobacco Institute commissioned a legal analysis from Covington and Burling on congress' exclusive authority to regulate tobacco products [140], and an analysis from Price Waterhouse on the tax revenue losses that would result from a cigarette sales ban [131–133].<br>• Winthrop, MA: Tobacco company consultants provided retailers with statistics regarding retail sales linked to cigarette sales [141] |
| | | Winthrop, MA (1997) | |
| | Create media/publicity campaigns | HI (1990) | • HI (1990): The Tobacco Institute prepared a press release entitled "Legislature considers tobacco prohibition; black market would thrive" [142] but it is unclear if it was released [143].<br>• Iroquois County, IL (1990): RJ Reynolds drafted "smoker's rights" letter to editor [144].<br>• Winthrop, MA (1997): Tobacco industry sent "instructional mailings" to residents and adopted slogan "Resist Prohibition!" [145]. |
| | | Iroquois County, IL (1990) | |
| | | Winthrop, MA (1997) | |

Abbreviations: HI Hawaii; IL Illinois; MA Massachusetts; ME Maine.

support varied, and included the likely positive impact of a sales ban on public health, the environment, health care costs, smoking cessation, youth smoking initiation, and smoking denormalization (Table 4). Many of these arguments were drawn from the growing literature showing an association between tobacco retailer density and smoking initiation, continued use, failed quit attempts, and relapse after cessation [20, 173–180]

## Discussion

From the earliest days of the modern US tobacco control movement, towns, cities and states have fielded proposals from state legislators, public health commissioners, health departments, medical societies, city council members, boards of health, and citizens seeking to end sales of cigarettes or tobacco products. Massachusetts was the most active, at both the local and state

**Table 3. Arguments against proposals to ban cigarette/tobacco sales made by the tobacco industry or its allies.**

| Argument | Where (when) employed |
|---|---|
| Societal costs | |
| • Lack of legal authority to ban [127, 150] | HI (1990) |
| | Salem, MA (1994) |
| • Legal product (no other legal product banned) [128, 146, 150–152] | HI (1990) |
| | Winthrop, MA (1997) |
| | Manhattan Beach, CA (2020) |
| • Slippery slope [150] | HI (1990) |
| • Infringes on adult choice/freedom [2, 66, 150, 153, 154] | HI (1990) |
| | Winthrop, MA (1997) |
| | Westminster, MA (2014) |
| | Manhattan Beach (2020) |
| • Proposal is radical, extreme, or unprecedented [125, 130, 153, 155, 156] | MA (1983) |
| | HI (1990) |
| | Winthrop, MA (1997) |
| | Westminster, MA (2014) |
| Economic costs | |
| • Loss of government tax revenue [58, 127, 150, 153, 157, 158] | HI (1990) |
| | ND (1990) |
| | Salem, MA (1994) |
| | Elk Grove Village, IL (2006) |
| • Will provoke inter-state [150] or "inter-city and inter-county animosities" [127] | HI (1990) |
| | Salem, MA (1994) |
| Costs to the tobacco industry | |
| • Retailers would suffer job/revenue losses [2, 112, 127, 128, 153, 154, 159]<br>  ○ Small businesses would be particularly impacted, as tobacco purchases lead to other purchases [117, 139, 141] | HI (1990) |
| | Salem, MA (1994) |
| | Winthrop, MA (1997) |
| | Westminster, MA (2014) |
| | Beverly Hills, CA (2019) |
| | Hermosa Beach, CA (2019) |
| | Manhattan Beach, CA (2020) |
| • Will foster antagonism against store clerks by irate customers [112] | Beverly Hills, CA (2019) |
| Law enforcement costs | |
| • Akin to alcohol prohibition: will create black markets, crime [117, 127, 150, 153, 157, 160, 161] | HI (1990, 2019) |
| | North Dakota (1990) |
| | Salem, MA (1994) |
| | Winthrop, MA (1997) |
| | Hermosa Beach, CA (2019) |
| Denial of intended public health benefits | |
| • Won't work (e.g., customers will go elsewhere while retailers suffer; youth will turn to black market, or get cigarettes from siblings and friends [2, 112, 127, 154, 159, 161] | Salem, MA (1994) |
| | Westminster, MA (2014) |
| | HI (2019) |
| | Beverly Hills, CA (2019) |
| | Manhattan Beach, CA (2020) |

*(Continued)*

**Table 3.** (Continued)

| Argument | Where (when) employed |
|---|---|
| • Lack of public support [66, 162] | Winthrop, MA (1997) |

Abbreviations: CA California; HI Hawaii; IL Illinois; MA Massachusetts; ND North Dakota.

level, but several other jurisdictions also made multiple attempts over the years to end tobacco sales. California, despite its history of strong tobacco control measures, was a late entrant to the field, with 5 sales ban proposals occurring only recently. Nonetheless, it was home to the only proposals to succeed.

Proposals were typically justified in the name of promoting health; more specific potential outcomes included reducing youth smoking initiation, promoting smoking cessation, and, occasionally, reducing tobacco-related healthcare costs. Despite the somewhat novel and arguably dramatic nature of these proposals, the rationales given were not unique to sales bans and were usually not made through emotional appeals. Bans on flavored tobacco products [181] and plain packaging of cigarettes [182], for example, have also been advanced using similar arguments.

Most often, proposed sales ban policies encompassed *all* tobacco products, although it was unclear why some jurisdictions occasionally deviated from this trend, proposing to end cigarette sales only. In 2019 and 2020, the scope of these policies broadened further, to include electronic cigarettes. At the same time, public health groups began offering more consistent support for these policies, suggesting that their support did not depend on maintaining electronic cigarettes as an alternative nicotine product when other tobacco products were banned. Indeed, their critique of Hawaii's 2019 proposal as too narrowly focused on combustible cigarettes suggests that future sales ban proposals with a broader scope may garner more support from public health organizations.

Tobacco companies relied on few strategies and tactics to oppose sales ban proposals, and, despite the extreme threat to their business model that such proposals might portend, employed none that have not been previously identified in analyses of tobacco industry campaigns against other tobacco control measures [32–34]. After a relatively public, multi-faceted campaign in Hawaii in 1990, the tobacco industry now appears to operate mostly behind the scenes, relying on allies to speak for it; this follows a pattern evident in its approach to other tobacco control efforts in the US, including the fight for clean indoor air in the 1990s, necessitated by its declining reputation [183]. A higher profile tactic that has not yet been employed–litigation–may be reserved until sales ban proposals in Beverly Hills and Manhattan Beach go into effect in January 2021.

Tobacco companies also used no new arguments to oppose tobacco sales ban proposals, advancing instead arguments drawing on themes previously identified in the literature [32]. While those broad themes might allow for the development of specific, novel argumentation against tobacco sales bans, we saw no evidence of that. Even the most likely candidate, a reference to alcohol prohibition in the US, was not entirely novel: the tobacco industry has for years positioned public health advocates as secret "prohibitionists" and has regarded exposing this secret agenda as an effective strategy to thwart tobacco control [184]. Although the industry deployed the Prohibition argument as recently as 2019 in Hawaii and Hermosa Beach, it may not be a reliably effective scare tactic. In San Francisco, California in 2018, for example, RJ Reynolds' ad campaign linking the city's flavored tobacco sales ban to alcohol prohibition–"Bans don't work. . . . Stop the Prohibition Proposition [185]–failed to sway voters, who upheld the ordinance in a referendum by a large margin, with 68% voting in favor [186].

**Table 4. Public health organizations' positions, rationales, and activities related to proposals to ban cigarette/tobacco sales.**

| Year | Location/ Bill Number | Public Health Organization(s) | Position | Rationale/arguments | Activities |
|------|------|------|------|------|------|
| 1983 | Massachusetts | • Action on Smoking and Health (ASH)<br>• Group Against Smoking Pollution (GASP) | ASH and GASP: No comment [163] | According to the Tobacco Institute, "no one wanted to jump on this bandwagon" [163] | N/A |
| 1997 | Winthrop, MA | • State Tobacco Control Program (MA TCP)<br>• MA American Cancer Society (ACS) | MA TCP: Opposed<br><br>MA ACS: Supported | MA TCP: Not enough scientific evidence to support policy; focus should be on developing less harmful sources of nicotine [160]<br><br>MA ACS: "It's a social experiment the whole country will be watching. More power to them" [155]. | Unknown |
| 2003 | North Dakota<br>HB 1174 | • American Heart Association<br>• American Lung Association<br>• North Dakota Medical Association<br>• North Dakota Public Health Association<br>• Tobacco Prevention and Control division of state Health Department | Opposed | • No evidence that a ban would prevent tobacco use [88, 164]; should focus instead on proven strategies [90]<br>• Lack of public support; would fail in similar manner as alcohol prohibition [164]<br>• Would jeopardize federal grant monies used for tobacco control [164] | Testified at House Finance and Taxation Committee hearing [90, 164] |
| 2006 | Elk Grove, IL | American Lung Association (ALA) | Not a policy priority, but would support if passed [165] | Will "support anything that will make the community smoke free and reduce the numbers of lung disease" [165]. | Two ALA representatives spoke at public meeting [165] |
| 2019 | Hawaii<br>HB 1509 | • State Department of Health (DOH)<br>• Coalition for Tobacco Free Hawaii (a program of the Hawaii Public Health Institute) | Opposed [although the Coalition supported the measure's intent] | • DOH & Coalition: Bill should include electronic cigarettes, cigars, and smokeless tobacco products<br>• Coalition: Legislature should consider evidence-based interventions, such as tobacco tax increases, clean indoor air policies, and full access to smoking cessation services, that have been proven to reduce adult smoking rates [161] | DOH & Coalition submitted written testimony to the House Committee on Health [161] |
| 2019 | Beverly Hills, CA | • Action on Smoking and Health (ASH)<br>• African American Tobacco Control Leadership Council<br>• American Academy of Pediatrics<br>• American Cancer Society<br>• American Cancer Society Cancer Action Network<br>• American Heart Association<br>• American Stroke Association<br>• Breathe LA<br>• Campaign for Tobacco Free Kids<br>• Cedar Sinai Medical Center<br>• Coalition for a Tobacco Free Los Angeles County<br>• National Stewardship Action Council<br>• Smokefree Air for Everyone (S.A.F.E.)<br>• Tobacco Education and Research Oversight Committee (TEROC)<br>• UCLA Fielding School of Public Health | Supported | • Tobacco sales inconsistent with human right to health<br>• Smoking is the leading cause of preventable death<br>• Will help smokers quit<br>• Will reduce youth smoking uptake and use<br>• Will reduce tobacco product-related litter and costs associated with clean up<br>• Will reduce future tobacco-related healthcare costs<br>• Other products harmful to public like asbestos and leaded gasoline have been banned [112, 166] | • Provided letters of support to city council<br>• Spoke at city council meetings [112, 166–169] |

(*Continued*)

**Table 4.** (Continued)

| Year | Location/ Bill Number | Public Health Organization (s) | Position | Rationale/arguments | Activities |
|------|-----------------------|--------------------------------|----------|---------------------|------------|
| 2019 | Hermosa Beach, CA | • Action on Smoking and Health (ASH)<br>• American Academy of Pediatrics<br>• Beach Cities Health District<br>• Surfrider Foundation | Supported | • Human rights duty to end tobacco sales<br>• Ban will support public health/send strong public health message<br>• Ban will protect youth<br>• Research shows that removing cigarettes from the marketplace results in fewer people smoking and reduces the likelihood of youth smoking initiation<br>• Ban will denormalize smoking<br>• Ban will decrease cigarette butt litter at beaches | • Spoke at city council meetings<br>• Provided written letters of support [115–117, 170] |
| 2020 | Manhattan Beach, CA | • American Academy of Pediatrics<br>• American Cancer Society Cancer Action Network<br>• American Heart Association<br>• American Lung Association<br>• Beach Cities Health District<br>• Surfrider Foundation | Supported | • Will reduce youth smoking initiation<br>• Will promote smoking cessation<br>• Will denormalize smoking<br>• Will protect health<br>• Will reduce tobacco-product related litter<br>• Will set precedent for other cities to follow | Spoke at city council meetings [121, 171, 172] |

Abbreviations: CA California; IL Illinois; MA Massachusetts.

The available evidence did not allow us to determine whether particular tobacco industry tactics or arguments were responsible for the failure of most previous tobacco sales ban proposals. We were able to identify, however, those that were used more often, which may serve as a proxy for those that the tobacco industry regarded as most successful. These included working collaboratively and mobilizing allies to make the claim that sales bans hurt retailers, particularly small businesses, and for no good reason, since bans were unlikely to reduce smoking. This approach fits neatly into the industry's overall discursive strategy in regards to any proposed tobacco control policy: exaggerating its potential costs and dismissing or denying its potential benefits [32].

Tobacco control advocates may take some comfort that tobacco sales bans do not appear to have stimulated a new set of tobacco industry tactics and arguments. Because those outlined here are familiar, advocates have, for the most part, experience countering them, often successfully. Advocates may also be reassured by the history, geography, and number of sales ban proposals put forward in the past 50 years, which may help minimize the sense that states and localities considering a ban are undertaking something unthinkable, unprecedented, or limited to places like California.

Until recently, public health organizations either did not support efforts to end tobacco sales, or did so only tepidly, because, they argued, sales bans lacked evidence of effectiveness. This approach was evident as recently as 2019 in Hawaii. However, in 2019 and 2020, numerous public health organizations supported Beverly Hills, Hermosa Beach, and Manhattan Beach in their efforts to end tobacco sales. In many cases, they based their support on the likely benefits a ban would have in reducing smoking prevalence among youth and adults, backed by research, developed over the past decade, exploring the link between tobacco outlet density and smoking behavior [20, 173–180]. Thus, while these public health organizations could not point to another US city that had adopted a tobacco sales ban and seen tobacco use fall, they had some evidence of a ban's likely effectiveness. (Hawaii's proposal to end cigarette sales over five years by gradually raising the minimum age of purchase, by contrast, had less of an evidentiary base to draw upon.)

Public health organizations' willingness to support a somewhat untested policy may have also been linked to the location of these proposals. California has low smoking prevalence and strong public support for interventions to reduce tobacco use and exposure to secondhand smoke, including 57% public support for a "gradual ban" on cigarette sales [187]. The state's Tobacco Control Program has adopted an "endgame" goal of ending the tobacco epidemic for all population groups by 2035 [188]. The bans proposed in California were also limited to individual cities, rather than the entire state, thus reducing the likelihood of possible negative outcomes, such as dramatic declines in state tax revenues, or the creation of a black market. In this environment, support for sales ban proposals was unlikely to be regarded as extreme or damaging to public health groups' reputations.

The support of public health organizations likely was not the only factor that altered the trajectory of sales ban proposals in 2019 and 2020. If this were the case, proposals would have succeeded not only in Beverly Hills and Manhattan Beach, but also in Hermosa Beach, since all three were supported by public health groups. One factor that may have set Hermosa Beach apart was greater concern about the impact of a ban on local businesses. Three city council members (of 5) linked their "no" vote to these concerns, versus one in Manhattan Beach and none in Beverly Hills (both of which had identified resources to help retailers transition). Thus, it appears that public health support is a necessary but not sufficient condition to aid passage of tobacco sales bans.

Going forward, other local governments in the US will likely consider adopting a tobacco sales ban. Communities seeking to adopt new tobacco control measures learn from one another [189]; indeed, this process has begun already in Southern California, with the mayor of Carson citing Beverly Hills as inspiration for his proposed ordinance to end tobacco sales, and a city council member in Hermosa Beach citing Manhattan Beach as inspiration. Recently announced prohibitions on the sale of flavored tobacco products at local [190] and state [191] levels may also inspire discussion of ending sales of *all* tobacco products in the interest of policy fairness and consistency; several jurisdictions in our study cited just such an interest when considering their own sales bans.

Our findings likely represent a conservative accounting of tobacco sales ban proposals, and responses to them, in the US in the past 50 years, as the sources we relied on to identify proposals to end sales are incomplete. For example, the tobacco industry documents archive contains only documents produced in the legal discovery process; among those documents subject to this process, some may have been destroyed, including those that might shed light on additional tobacco industry tactics and arguments [192]. Another limitation of the archive is that it contains fewer recent documents, further limiting our understanding of how the tobacco industry responded to recent tobacco sales ban proposals, and requiring us to rely for this information on media accounts and records of public comments. The large size of the archive also means that we may not have identified every relevant document, thus underestimating the number of tobacco sales ban proposals. Similarly, while the media database we relied on for additional information encompassed a wide range of news sources over time, there were fewer news sources indexed in the 1970s than in later decades, potentially leading to an undercount. Tobacco companies may have also failed to track every tobacco sales ban proposal, particularly at the local level. Archival material that did not always capture details of legislative debate and public comment, and limited media coverage of early proposals may also have resulted in an incomplete picture of public health organizations' responses.

## Conclusion

Ending the sale of tobacco products is not a new idea. Unsurprisingly, the tobacco industry has repeatedly opposed it, while somewhat surprisingly, public health groups have only

recently endorsed it. As the tobacco sales bans in Beverly Hills and Manhattan Beach come into effect in 2021, their impact will become clearer, providing valuable information to other jurisdictions considering their own approach to tobacco sales. They will also truly no longer be "unprecedented," but may be regarded as a logical next step in achieving an end to the tobacco epidemic.

## Supporting information

**S1 Data. Salem, Massachusetts Board of Health meeting minutes, April-September 1994.**
(PDF)

**S2 Data. Gray, Maine Town council meeting minutes, December 1, 1998.**
(PDF)

**S3 Data. Westminster, Massachusetts Board of Health meeting minutes, May 21, 2014.**
(PDF)

## Acknowledgments

The authors would like to thank Elizabeth A. Smith for reading and commenting on an early version of the manuscript.

## Author Contributions

**Conceptualization:** Patricia A. McDaniel, Ruth E. Malone.

**Data curation:** Patricia A. McDaniel.

**Formal analysis:** Patricia A. McDaniel, Ruth E. Malone.

**Funding acquisition:** Ruth E. Malone.

**Methodology:** Patricia A. McDaniel.

**Project administration:** Ruth E. Malone.

**Resources:** Ruth E. Malone.

**Supervision:** Ruth E. Malone.

**Writing – original draft:** Patricia A. McDaniel.

**Writing – review & editing:** Patricia A. McDaniel, Ruth E. Malone.

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
