## [Decision Letter · Decision Letter 0]

18 Feb 2020

PONE-D-19-27001

Tobacco industry and public health responses to state and local efforts to end tobacco sales from 1969-2019

PLOS ONE

Dear Dr. McDaniel,

Thank you for submitting your manuscript to PLOS ONE. After careful consideration, we feel that it has merit but does not fully meet PLOS ONE’s publication criteria as it currently stands. Therefore, we invite you to submit a revised version of the manuscript that addresses the points raised during the review process.

There are two things that I think are especially important for a revision (also highlighted by the referee reports). First, there needs to be more information on the search strategy and what was actually done. Second, the authors need to do a lot more on the analysis of the data -- what is it about this instance that the ban went through while in the others it did not? What kinds of initiatives by the state draws what kinds of an industry response? Are there correlations in the characteristics that help one understand the strategies adopted by the industry. More statistical rigor will help in making a stronger case to understand the specific question posed.

We would appreciate receiving your revised manuscript by Apr 03 2020 11:59PM. To enhance the reproducibility of your results, we recommend that if applicable you deposit your laboratory protocols in protocols.io, where a protocol can be assigned its own identifier (DOI) such that it can be cited independently in the future. For instructions see: http://journals.plos.org/plosone/s/submission-guidelines#loc-laboratory-protocols

We look forward to receiving your revised manuscript.

Kind regards,

Renuka Sane

Academic Editor

PLOS ONE

Journal Requirements:

"I have read the journal's policy and the authors of this manuscript have the following competing interests:

PAM: Personal financial interests: Since 2006, I have been a full-time faculty employee of the University of California, San Francisco. My salary is provided by funds from research grants. I have received honoraria from the U.S. Department of Justice (for serving as a tobacco industry documents consultant for United States of America vs. Philip Morris, et al.), and Cancer Research UK (for preparing a report on views of the idea of a tobacco “endgame”). I have participated in tobacco control advocacy.

Organizational financial interests: Within the last 5 years I have received grant funding from the California Tobacco-Related Disease Research Program (research funds derived from the state tax on cigarettes), and have worked on projects funded by the National Cancer Institute and California Tobacco Control Program.

Interests of related parties: None to declare.

REM: Personal financial interests: Since 1997, I have been a full-time faculty employee of the University of California, San Francisco. My salary is provided by funds from the state of California and my research grants. I have received travel/accommodation expenses and consulting fees or honoraria from the U.S. Department of Justice (for serving as a tobacco industry documents consultant for United States of America vs. Philip Morris, et al.), World Health Organization (for serving on the Expert Panel on Tobacco Industry Interference with Tobacco Control), American Legacy Foundation (for serving on an award selection panel), Clearway Minnesota (for serving as a grant proposal reviewer), U.S. Centers for Disease Control (for consulting on a tobacco industry documents research project), NIH (for serving as a grant proposal reviewer) and Cancer Research UK (for preparing a report on views of the idea of a tobacco “endgame”). I own one share each of Philip Morris International, Reynolds American, and Altria stock for research and advocacy purposes and have participated in tobacco control advocacy. I receive an annual honorarium and reimbursement of travel/accommodation expenses from BMJ Publishing Group Ltd (for work as editor-in-chief of Tobacco Control). I have also received travel/accommodation expenses and honoraria for speaking to various public health groups. In addition, I have received funding for reviewing documents as a potential expert witness for plaintiffs’ legal cases involving tobacco industry activities.

Organisational financial interests: Within the last 5 years I have received grant funding from the National Cancer Institute, National Institutes of Health, the California Tobacco-Related Disease Research Program (research funds derived from the state tax on cigarettes), and the California Tobacco Control Program.

Non-financial interests: I have published or collaborated on research with more than 50 colleagues, postdoctoral fellows and students. I recuse myself from handling or reviewing papers submitted by these colleagues and others from my institution (UCSF).

Interests of related parties: None to declare

Both authors provided written information in support of the Beverly Hills 2019 ordinance to end tobacco sales."

Reviewers' comments:

Reviewer's Responses to Questions

**Comments to the Author**

1. Is the manuscript technically sound, and do the data support the conclusions?

Reviewer #1: Partly

Reviewer #2: Yes

2. Has the statistical analysis been performed appropriately and rigorously? 

Reviewer #1: N/A

Reviewer #2: N/A

3. Have the authors made all data underlying the findings in their manuscript fully available?

Reviewer #1: Yes

Reviewer #2: Yes

4. Is the manuscript presented in an intelligible fashion and written in standard English?

Reviewer #1: Yes

Reviewer #2: Yes

5. Review Comments to the Author

Reviewer #1: Main Comments

Thank you for the opportunity to review this paper which analyses efforts, over many decades, to adopt bans on the sale of cigarettes in the US, and industry efforts to oppose such bans. Fuller understanding of tobacco industry efforts to oppose, undermine and even prevent the adoption of effective tobacco control measures in different policy contexts is an important knowledge contribution. On this basis, the aim of this article is important. However, I am not convinced of the extent to which this aim has been achieved by the authors in two respects. First, the authors state that there has been no detailed analysis to date of how the industry “has anticipated and reacted to efforts to end tobacco sales.” This is a statement of the originality of the subject matter and I would have liked to see a tight brief review first of what we already do know about industry opposition to tobacco control measures in the US, which measures, what their tactics have been, and with what effect. This would provide important context for the analysis of sales bans that follows. Second, the authors need to explain the significance of understanding industry strategies regarding sales bans beyond that it has not been studied before. Is this to confirm more of the same, regarding industry activities, or is there something distinctly valuable about how the industry has behaved regarding sales bans that extends our understanding of industry activities and strategies?

The other main comment that I have about this paper is that the balance between description and analysis needs to shift more towards the latter. While the material presented could make a potentially important contribution to knowledge, as mentioned, the paper is currently largely a description of the information found in internal documents and media sources about what bans were proposed, when and where, and what the industry said to oppose them. There is need for tighter analysis which draws this information together and gives it higher meaning. This would be helped by providing a clearly stated research question at the beginning beyond finding out what the industry has done to oppose sales bans. Can the authors explain the timing of the sales ban policy proposals, how they subsequently faired and why, why the Beverly Hills policy was adopted, and the prospects for future adoption of sales bans as an effective tobacco control measure. This might not all be possible, given the quality of the data available, but delving down more deeply into selected policies is needed to give this paper more analytical substance.

Specific Comments

• The Abstract could be revised to sharpen its statement of the key contribution to knowledge of this paper (originality and significance). It could set out, in particular, not only what the paper has done but why it has done this. Is there a specific research question you are seeking to answer? The purpose of the paper is also not clearly stated. Overall, what does the paper contribute as new knowledge of the tobacco industry and its opposition to tobacco control policies, and why is this important to know?

• It could be made clearer whether the sales bans proposed/adopted refer to all tobacco/nicotine products or only cigarettes and why. What is the scope of the various policies put forth/adopted, and to what extent does this vary across jurisdictions and time? What have been the key debates surrounding bans? This would set up the subsequent analysis more tightly.

• In the Methods, the first sentence may no longer be needed. The substantial number of publications using internal industry documents from this source might make it sufficient to simply name the UCSF Truth Documents collection. The full name of the depository, with a link to it, may just be needed.

• More detail is needed in the Methods about how the documents were searched, all the keyword terms used, what the exclusion/inclusion criteria applied were, and how the documents identified were then analysed (e.g. thematically coded). The document analysis is too sparsely described and needs to provide sufficient detail to enable results to be reproducible.

• Add “in the US” to end of sentence 1 of Results. There is a universal tone adopted in the paper and, given the global readership of this journal, it is important to acknowledge that this paper is only about the US context.

• The tobacco industry is described in quite broad terms and referred to as the “industry”. More detailed description, of which companies or industry-related body acted when and where, to oppose the various sales bans, is needed.

• The extent to which industry opposition was a decisive factor in the failure to adopt bans is not clear from the paper’s analysis. This relates to my main comments about description versus analysis. There is no attempt to explain, with evidence, what happened. Some speculation in the Discussion but, to strengthen this paper, this analysis needs to be more central to the research.

• The statement on p. 15, line 153-154, that the industry “consistently mobilized or prepared to mobilize allies and community members to oppose tobacco sales bans” does not seem supported by the evidence presented in the paper. There is a lack of detail about industry activities, beyond the framing of the various proposed policies. Was there an industry strategy across companies for example like Project Whitecoat? What allies were mobilized and what did they do? To what extent was opposition in different cities/towns/states coordinated within the industry? Did industry arguments change over time or in nature? Which were more effective and why? I realize that internal documents may not provide much information but this is needed if one is to describe industry efforts as somehow organized in this way.

• The authors describe the adoption of a sales ban in Beverly Hills as a “turning point” in 2019. I wonder if this is premature given that it is only one municipality, no others have so far followed suit, and it may be an aberration especially given its unrepresentative citizenry. Traditionally, the US public has been averse to prohibitions and tend to lean towards libertarian values. The failure to ban guns, for example, is illustrative of this. What is in the pipeline? Are other sales bans being proposed across the state or US?

• There are some passing comparisons of industry arguments with other public policy issues but this is not done systematically. Might comparative analysis allow the authors to draw wider conclusions or lessons going forward?

• If the authors can make the case that Beverly Hills is,indeed, a turning point, I wondered if this could form the core narrative in this paper to give it more analytical substance. If so, the paper needs to explain more specifically what changed. Was it industry opposition or public health efforts? Or something else. It would be interesting to discuss, in this context, what prevented adoption previously? I would guess that the rapid uptake of e-cigarettes has been a key factor. This is mentioned in passing in the Discussion. Historically, the public health community has not supported sales bans because of the highly addictive nature of nicotine. A ban would have a serious impact on a large number of existing tobacco users. The widespread availability of e-cigarettes creates an alternative (not a safe one albeit) which seems to have softened opposition to banning combustible tobacco products. Given thousands of cases of acute vaping-related lung disease, it would now be interesting to see if support for sales bans wanes.

• Relatedly, the explanation in the Discussion of the adoption of the Beverly Hills ban, that it “may have been related, in part, to the support of public health groups”, seems a bit broad and speculative. Indeed, this is what one might expect to have answered more clearly in this paper. Rather than the Discussion stating “it could be this” or “it could be that”, the paper needs to pull together the findings more concertedly and state the “so what” takeaway messages from the evidence reviewed.

• How effective is the ban in Beverly Hills when it is possible to go to a neighbouring municipality to purchase tobacco products? Or internet sales? What can we learn from sales bans regarding supply-side tobacco control measures?

• The main results of the paper seem to be that the industry misrepresented bans as unprecedented and framed them in specific terms. The Discussion picks up this point again. This does seem a bit weak and underwhelming as the key finding arising from the research. There is copious evidence that the industry has often not told the truth. What would be far more compelling, as a contribution to knowledge, is details about activities, strategies and tactics. How differently or not did the industry behave on sales bans? How effective was the industry? What counterarguments are effective to challenge the industry’s efforts?

Reviewer #2: Review:

Introduction

- Line 49 – perhaps it is worth adding the term ‘licit’ or ‘legal’ in the sentence “… the single most deadly [licit/legal] consumer product.” The fact that tobacco is legal seems to be the punctuation to this point.

- Line 50 – I think this is a clear and accurate point pertaining to the health implications of tobacco leading to considerations of banning cigarettes. I wonder if it is worth adding that this movement is also tied up in competing/conflicting values of individualism, hygiene (the puritan movement), the proper place of government in society, etc. This is a minor point, but may cast this history in a broader context beyond simply a technical and linear relationship between health risk and public policy.

- Very clear and concise introduction to the topic of the paper. Compelling points about the need to examine the industry response to proposals and actions to ban the sale of tobacco.

Methods

- I think it will be important to provide more information on the search strategy to retrieve the 110 relevant documents. How did the authors ensure that all relevant documents were identified? Do the authors think that it is possible that any documents were missed? If so, why, and with what implications for the analysis?

- The news media search strategy is well described and provides a clear picture of how the authors approached this search. The strategy of identifying cities and states where tobacco bans were brought forward seems like it could be utilized in the industry documents as well, using the cities and states as search terms in combination with ‘sales ban’ and any other relevant substantive terms. Did the authors utilize this approach in the search strategy for the industry documents?

- Lines 109-111 – please provide more information on the number of ‘selected key documents’ reviewed by the second author? How were these documents selected? How was disagreement resolved? And prior to this, what was the second author examining in these documents? What type of verification was sought? Finally, please explain what is meant by ‘case study’? Perhaps this could also be clarified by the presentation of a more pointed research question in the introduction section. In other words, was this a case study of industry responses, industry action, how opposition was mobilized, who was engaged in this opposition, what types of arguments were brought forward by industry opponents, etc.?

Results

- Line 134 – the authors note that the industry ‘apparently prepared a 6-page document entitled …’ Is there a reason for the term ‘apparently’? Were the authors unable to obtain this document? If not, I think it would be beneficial to mention how the knowledge of this document was gained.

- The results are quite interesting. The two tables are helpful to gain an overview of the key proposal and the specific opposition to these proposals of banning tobacco. As I was reading the results I kept thinking about who it was bringing forward the proposal to ban tobacco in the various localities. As the authors find, the tobacco control community wasn’t too engaged in supporting or opposing these proposals. It would enrich the findings to add some discussion of the motivations or stated rationale by those proposing a ban on tobacco. Without this context the paper does provide interesting insights into the way that the industry opposed such measures, however it is quite similar to opposition to other tobacco control measures. The novelty is perhaps in the messaging of such a ban being ‘radical’ or ‘fanatical’, although even this type of messaging is seen in other novel measures such as standardized tobacco packaging or flavouring bans. E.g.

Lencucha, R., & de Lima Pontes, C. (2018). The context and quality of evidence used by tobacco interests to oppose ANVISA’s 2012 regulations in Brazil. Global public health, 13(9), 1204-1215.

Lencucha, R., Drope, J., & Labonte, R. (2016). Rhetoric and the law, or the law of rhetoric: how countries oppose novel tobacco control measures at the World Trade Organization. Social Science & Medicine, 164, 100-107.

Among others

Discussion

- The discussion is well-written and compelling. My only suggestion is that the authors go a bit deeper into the hypothesis that the involvement of tobacco control-public health organizations may have been the factor that led to the passing of the sales ban in Beverly Hills. Is there any evidence of this involvement tipping the scales in the process of adopting the legislation?

- Similar to my final comment on the results section, I think it is important for the authors to add more information about the way that that tobacco ban was framed and argued for by those bringing forward the proposal. It is both academically interesting and practically important to know how such a novel bill was framing and supported both in terms of evidence and argumentation. I would like to see this added to the discussion.

Overall, this is an interesting, timely and well-research paper. I think it will clearly contribute to the literature on industry approaches to novel tobacco control measures.

6. PLOS authors have the option to publish the peer review history of their article (what does this mean?). If published, this will include your full peer review and any attached files.

Reviewer #1: Yes: Kelley Lee

Reviewer #2: No

---

## [Editor Report · Decision Letter 1]

6 May 2020

Tobacco industry and public health responses to state and local efforts to end tobacco sales from 1969-2020

PONE-D-19-27001R1

Dear Dr. McDaniel,

We are pleased to inform you that your manuscript has been judged scientifically suitable for publication and will be formally accepted for publication once it complies with all outstanding technical requirements.

With kind regards,

Renuka Sane

Academic Editor

PLOS ONE
---

## [Editor Report · Acceptance letter]

13 May 2020

PONE-D-19-27001R1 

Tobacco industry and public health responses to state and local efforts to end tobacco sales from 1969-2020 

Dear Dr. McDaniel:

I am pleased to inform you that your manuscript has been deemed suitable for publication in PLOS ONE. Congratulations! Your manuscript is now with our production department. 

With kind regards,

on behalf of

Dr. Renuka Sane 

Academic Editor

PLOS ONE